# ROMA: Regularization for Out-of-distribution Detection with Masked Autoencoders

## Abstract

Existing out-of-distribution (OOD) detection methods without outlier exposure learn effective in-distribution (ID) representations distinguishable for OOD samples, which have shown promising performance on many OOD detection tasks. However, we find a performance degradation in some challenging OOD detection, where pre-trained networks tend to perform worse during the fine-tuning process, exhibiting the over-fitting of ID representations. Motivated by this observation, we propose a critical task of hidden OOD detection, wherein ID representations provide limited or even counterproductive assistance in identifying hidden OOD data. To address this issue, we introduce a novel Regularization framework for OOD detection with Masked Autoencoders (ROMA), which utilizes the masked image modeling task to regularize the network. With distribution-agnostic auxiliary data exposure, ROMA notably surpasses previous OOD detection methods in hidden OOD detection. Moreover, the robustness of ROMA is further evidenced by its state-of-the-art performance on benchmarks for other challenging OOD detection tasks.

## 1 Introduction

Neural networks are unreliable to provide predictions for samples that fall outside of the training distribution (in-distribution), leading to insecurity of deep learning in safety-critical applications (Zhu et al., 2023; Caruana et al., 2015; Eykholt et al., 2018). Thus, it is important to enable the neural network to effectively distinguish such out-of-distribution (OOD) inputs from in-distribution (ID) samples, which is the task of OOD detection. There is a rich line of research on OOD detection in recent years, which can be roughly divided into two categories: 1) *Data-driven methods* utilize outlier exposure to let the network learn OOD features in a supervised way (Hendrycks et al., 2019b; Mohseni et al., 2020), which require human effort to determine the distribution of outliers (Katz-Samuels et al., 2022). 2) *Training-driven methods* are based on the source data without outliers. These methods use source data to train or fine-tune the network for effective ID representation to distinguish OOD samples (Bendale & Boult, 2016; Hendrycks & Gimpel, 2016; Liang et al., 2018; Wei et al., 2022). Due to additional human effort (Katz-Samuels et al., 2022), data-driven methods have limited application in the real world and the latter ones attract more attention from researchers. Although existing training-driven methods achieving good performance of OOD detection (Hsu et al., 2020; Liu et al., 2020; Wang et al., 2022a), recent research discovers that there is significant performance degradation in challenging OOD detection (Yang et al., 2022; Zhang et al., 2023b), indicating the existence of specific OOD samples to trick the network with effective ID representation.

In our work, we address a novel OOD detection task (named hidden OOD detection) in the context of image classification, wherein the ID representation offers limited or even counterproductive assistance for OOD detection. Specifically, hidden OOD samples in this task exhibit a distinct distribution from the ID samples in a generalized feature space but share similar features in the network's feature space learned for ID representation. As ID representation is no longer effective to detect hidden OOD samples, learning for ID objective makes it easier for those OOD samples to trick the network, which poses a potential risk to training-driven OOD detection methods and constrains their upper bounds for OOD detection.

To address this prospective over-fitting concern in OOD detection, we construct specific (ID, hidden OOD) pairs and propose a novel Regularization framework for OOD detection with Masked Autoencoders (ROMA). With great success of pre-trained masked autoencoders (MAE) (He et al., 2022a;

Feichtenhofer et al., 2022; Bai et al., 2023), the MAE with masked image modeling (MIM) (He et al., 2022a; Bao et al., 2021) on large dataset extracts generalized features beneficial to numerous downstream tasks. Inspired by this, ROMA adopts MAE pre-trained by MIM pretext task as the basic network. During the fine-tuning process, ROMA enables the MAE to learn supervised classification from source images and self-supervised reconstruction from masked auxiliary images simultaneously, regularizing the network to preserve generalized features against hidden OOD samples. It is noteworthy that auxiliary images in self-supervised reconstruction are totally unlabeled. Unlike methods based on outliers mining, unlabeled data used for ROMA can be easily sampled from the pre-training datasets or even the source data.

We extensively evaluate ROMA in hidden OOD detection and standard OOD detection benchmarks based on OpenOOD (Yang et al., 2022). ROMA shows effective regularization for hidden OOD detection and improves the metrics (AUROC↑/FPR95↓) of OOD detection from 91.80%/33.88% to 96.91%/19.92%. For benchmarks in OpenOOD, ROMA outperforms competitive baselines, achieving state-of-the-art (SOTA) AUROC of 97.32%, 89.81%, and 79.23% in the challenging tasks of near-OOD detection for CIFAR-10, CIFAR-100,and ImageNet-1K, respectively. Moreover, we conduct benchmarks in OOD detection with uneven training distribution, where the robustness of ROMA is verified by its minimal performance degradation. In summary, the contribution of this paper is three-fold:

- We introduce a hidden OOD detection, representing a unique scenario where the ID representation learned from the source data offers limited or even counterproductive assistance in detecting certain OOD samples.

- We propose a unified regularization framework ROMA, which utilizes masked image modeling to regularize the network with both effective ID representation and generalized features for OOD detection.

- Extensive experiments demonstrate that ROMA not only effectively implements hidden-OOD detection but also achieves SOTA performance in standard and other challenging OOD detection benchmarks.

## 2 RELATED WORK

**Masked Autoencoders.** Motivated by masked language modeling that is highly successful in NLP (Kenton & Toutanova, 2019; Brown et al., 2020), MAE (He et al., 2022a) is introduced as a neural network pre-training framework that reconstructs the original image from its partial observation. The pre-trained encoder of MAE can be fine-tuned in numerous downstream tasks with excellent performance (Cai et al., 2023; Hess et al., 2023; Bachmann et al., 2022), which shows a broad impact on visual recognition. As pre-trained networks have shown superiority in OOD detection (Li et al., 2023; Hendrycks et al., 2019c; Fort et al., 2021), it is reasonable for us to apply MAE, a more effective pre-training framework, for OOD detection. Since MAE is totally unsupervised, ROMA based on MAE frees itself from dependence on labeled auxiliary dataset, which is crucial for the practical deployment.

**Challenging OOD Detection.** Current OOD detection lacks generalized benchmarks (Yang et al., 2022) and most studies simply select publicly available classification datasets as ID or OOD samples. These works can easily lead to a significant difference in the distribution of ID and OOD samples, where most methods perform remarkably well. However, good performance in such benchmarks does not imply effective OOD detection, given the complexity of real-world scenarios. Recent research starts to focus on this issue and proposes more challenging OOD detection. For instance, near-OOD detection (Yang et al., 2022; Park et al., 2023) addresses the presence of OOD samples with finer-grained semantic shifts; full-spectrum OOD detection (Yang et al., 2023; Zhang et al., 2023b) is proposed to consider ID samples with covariate shifts; OOD detection with long-tailed recognition (Wang et al., 2022b; Li et al., 2022) accounts for class imbalance in ID samples. In a similar vein, our work introduces hidden OOD detection and examines the existence of certain OOD samples in a generalized feature space.

**OOD Detection with Auxiliary datasets.** OOD detection with outliers is effective but controversial. Initial works learn features for distinguishing ID and OOD samples (Hendrycks & Gimpel, 2016; Liu et al., 2020) based on outliers directly sampled from the distribution of the target OOD samples.

However, the distribution of the target OOD samples is commonly unknown. In this regard, some methods explore leveraging information from natural outliers that do not rely on prior information of the target OOD samples (Hendrycks et al., 2019b; Mohseni et al., 2020; Chen et al., 2021; Zhang et al., 2023a), where diverse data are abundantly available. Although these methods no longer require the outliers with specific distribution, they still need human effort to ensure distinct distributions between the outliers and ID samples (Katz-Samuels et al., 2022). Moreover, inappropriate outliers can lead to adverse effects on OOD detection. There are also other works tending to generate virtual outliers (Du et al., 2021; He et al., 2022b; Narayanaswamy et al., 2023). However, it is hard to ensure the effectiveness of the features learned from these virtual outliers, as the features of virtual outliers may significantly deviate from those of natural samples. In contrast to outliers, the auxiliary data in ROMA does not require any outlier mining and is easily sampled from natural images, which is promising for practical applications.

## 3 PRELIMINARIES

In this work, we consider OOD detection in the context of (supervised) image classification. The problem statement considering both standard OOD detection and hidden OOD detection is described in this section.

### 3.1 STANDARD OOD DETECTION

OOD detection can be formulated as a binary classification problem. The input space and target space are denoted by $\mathcal{X}$ and $\mathcal{Y} = \{1, 2, ..., K\}$, where $K$ is the number of ID classes. The training dataset $\mathcal{D}_{in} = \{(\mathbf{x}_i, y_i) | \mathbf{x}_i \in \mathcal{X}, y_i \in \mathcal{Y}, i \in [1, N]\}$ is sampled from the in-distribution $p_{in}(\mathbf{x}, y)$, where $N$ is the size of $\mathcal{D}_{in}$. Let $\mathcal{P}_{in}$ denote the marginal distribution of $p_{in}(\mathbf{x}, y)$ on $\mathcal{X}$. Thus, the goal of OOD detection is to infer whether the given input $\mathbf{x}_{test} \in \mathcal{X}$ is sampled from the $\mathcal{P}_{in}$ or not ($\mathcal{P}_{out}$). $\mathcal{P}_{out}$ can be any distribution that does not overlap with $\mathcal{P}_{in}$. Methods for detecting $\mathcal{D}_{out} \sim \mathcal{P}_{out}$ focus on building an effective detector $G(\cdot)$ based on networks, of which the decision is made via a threshold comparison:

$$G(\mathbf{x}) = \text{ID, if } S(\mathbf{x}) \geq \gamma; \text{otherwise}, \ G(\mathbf{x}) = \text{OOD},$$

where samples with lower confidence scores $S(\mathbf{x})$ are classified as OOD and $\gamma$ is the threshold. As OOD detection is expected to accurately identify OOD samples without affecting the performance of the original task, the ID classification accuracy of $\mathbf{x}_{in} \sim \mathcal{P}_{in}$ is also important for benchmarks (Yang et al., 2022). For datesets of OOD detection, standard benchmarks utilize different public classification datasets to form (ID, OOD) pairs like (CIFAR-10, CIFAR-100 (Krizhevsky & Hinton, 2009)). These (ID, OOD) pairs need to meet the requirement that datasets for ID and OOD do not share the same categories.

### 3.2 HIDDEN OOD DETECTION

The architecture of an classification network can be understood as consisting of an encoder and a decoder, where the encoder $Encoder(\cdot)$ is used to learn features for distinguishing $\mathbf{x}_{in}$ from different categories and the decoder is a linear $Classifier(\cdot)$. $Encoder(\cdot)$ transfers the input $\mathbf{x}$ into the feature space $\mathcal{H}$, which is then mapped into the target space $\mathcal{Y}$ by $Classifier(\cdot)$. Specifically, the encoder generates the corresponding feature $\mathbf{h} = Encoder(\mathbf{x})$, where $\mathbf{h} \in \mathbb{R}^d$ and $d$ is the dimension of $\mathbf{h}$. The classifier is employed to predict the label of $\mathbf{x}$ according to $\mathbf{h}$:

$$logits = \mathbf{W}^T \mathbf{h} + \mathbf{b}, \ p(y|\mathbf{x}) = Softmax(logits), \ \widehat{y} = \underset{k}{\operatorname{argmax}} \ p(y = k|\mathbf{x}),$$

where $\mathbf{W} \in \mathbb{R}^{d \times K}$ and $\mathbf{b} \in \mathbb{R}^K$ are the weight and bias of the classifier, respectively. $logits \in \mathbb{R}^K$ represents the energy of each class and the higher energy corresponds to the fact that more effective features representing a specific class from $\mathcal{H}$ can be extracted. $p(y|\mathbf{x})$ is the probability distribution obtained by Softmax function in target space $\mathcal{Y}$, and $\widehat{y}$ is the final prediction of the network.

Considering OOD detection, we use $\mathcal{P}_{out}^{\mathcal{H}}$ and $\mathcal{P}_{in,k}^{\mathcal{H}}$ to denote the distribution of $\mathbf{x}_{out} \sim \mathcal{P}_{out}$ in $\mathcal{H}$ and the distribution of ID samples belonging to the $k$-th class of $\mathbf{x}_{in} \sim \mathcal{P}_{in}$ in $\mathcal{H}$, respectively. For $\forall k \in \mathcal{Y}$, following the assumption that there is certain difference between $\mathbf{h}_{out} \sim \mathcal{P}_{out}^{\mathcal{H}}$ and $\mathbf{h}_{in,k} \sim \mathcal{P}_{in,k}^{\mathcal{H}}$, deep learning-driven OOD detection methods try to build effective $G(\cdot)$ based on $\mathbf{h}$,

*logits*, and $p(y|\mathbf{x})$. Learned from the ID classification task, the approximated feature space $\bar{\mathcal{H}}$ for ID representation is proven to be effective to distinguish OOD samples, which is considered as the core of OOD detection (Li et al., 2023; 2024). In current benchmarks for OOD detection, strong classification capabilities always lead to good performance in OOD detection (Vaze et al., 2021), while OOD samples hard to be distinguished by ID representation in $\bar{\mathcal{H}}$ are rarely noticed.

To highlight these OOD samples and thus address the phenomenon that good ID classification performance leads to poor OOD detection, we propose a task of hidden OOD detection focusing on specific OOD samples $\widehat{\mathbf{x}}_{out} \in \mathcal{D}_{out}$ whose $\widehat{\mathcal{P}}_{out}^{\mathcal{H}}$ will get close to $\mathcal{P}_{in,k}^{\mathcal{H}}$ during the learning process of $\bar{\mathcal{H}}$. Hence, the $\widehat{\mathbf{x}}_{out}$ satisfies

$$\lim_{\mathcal{H} \to \bar{\mathcal{H}}} G_{\mathcal{H}}(\widehat{\mathbf{x}}_{out}) = \text{ID}, \ s.t. \ G_{\widehat{\mathcal{H}}}(\widehat{\mathbf{x}}_{out}) = \text{OOD},$$

where $G_{\mathcal{H}}(\cdot)$ is the detector $G(\cdot)$ when the feature space of network is $\mathcal{H}$, and $\widehat{\mathcal{H}}$ represents a generalized feature space. For these OOD samples, features for ID representation have limited or even negative assistance, which make them easier to trick the network with ID supervised-learning. We use a toy example of (ID, hidden OOD) pair found in ImageNet-30 (Hendrycks et al., 2019c) to showcase the blind spot for ID representation (more examples are shown in Appendix A.4). Specifically, the (ID, hidden-OOD) example can be expressed as ({'Dragonfly','Tank'},{'Forklift'}) (Figure 1a), where $\mathcal{D}_{in}$ contains images labeled dragonfly and tank, and $\mathbf{x}_{out}$ is sampled from images labeled forklift. We apply a pre-trained $Encoder(\cdot)$ to fine-tune on $\mathcal{D}_{in}$, whose $\mathcal{H}$ will be changed from a generalized feature space $\widehat{\mathcal{H}}$ to a specific feature space $\bar{\mathcal{H}}$. As displayed in Figure 1b, the performance of OOD detection is worse as the training progresses. This over-fitting phenomenon indicates that most forklift images can be considered hidden-OOD in this case and the learning of the specific $\bar{\mathcal{H}}$ for ID representation will make the network blind to hidden OOD samples.

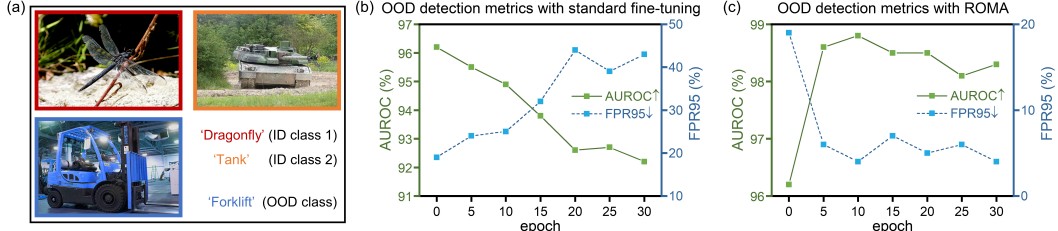

Figure 1: The performance (FPR95↓ and AUROC↑) of OOD detection in the fine-tuning process of the pre-trained network released by MAE on a (ID, hidden-OOD) pair (a), where (b) is fine-tuned with normal cross-entropy loss and (c) is fine-tuned with ROMA. The confidence score is based on Mahalanobis distance (Lee et al., 2018).

## 4 METHODS

In this section, we first introduce how to find samples from standard datasets for the benchmark of hidden OOD detection. We then elucidate the masked image modeling crucial for aiding hidden OOD detection and finally introduce the regularization framework ROMA.

### 4.1 HIDDEN OOD SAMPLES MINING

Labeled dataset $\mathbf{X}_{in}$ like CIFAR (Krizhevsky & Hinton, 2009) are widely used to benchmark OOD detection and we will show our method to construct hidden OOD dataset $\widehat{\mathbf{X}}_{out}$ from the corresponding OOD dataset $\mathbf{X}_{out}$. We assume that the pre-trained network has a generalized feature space $\widehat{\mathcal{H}}$ as the large-scale pre-training dataset $\mathbf{X}_P$ like ImageNet (Deng et al., 2009) is distributed widely beyond $\mathbf{X}_{in}$; the fine-tuned network has the feature space $\bar{\mathcal{H}}$ for ID representation as the recognition of $\mathbf{X}_{in}$ is the main objective. FPR represents the probability that the ID sample $\mathbf{x}_{in}$ is incorrectly identified as OOD by $G(\cdot)$. Therefore, we can calculate the corresponding threshold $\gamma_{\widehat{\mathcal{H}}}$ and $\gamma_{\bar{\mathcal{H}}}$ based on selected $\text{FPR}_{\widehat{\mathcal{H}}}$, $\text{FPR}_{\bar{\mathcal{H}}}$ and $S_{\widehat{\mathcal{H}}}(\cdot)$, $S_{\bar{\mathcal{H}}}(\cdot)$. Here, we use $S_{\mathcal{H}}(\cdot)$ based on Mahalanobis distance (Lee et al., 2018)

$$S_{\mathcal{H}}(\mathbf{x}) = -\min_{i} (\mathbf{h} - \boldsymbol{\mu}_i)^T \boldsymbol{\Sigma}^{-1} (\mathbf{h} - \boldsymbol{\mu}_i),$$

where $\mathbf{h}$ represents the feature of an input $\mathbf{x}$. $\boldsymbol{\mu}$ and $\boldsymbol{\Sigma}$ are the class mean representation and the covariance matrix derived from training data. It is noted that the training data is regarded as one category for $S_{\widehat{\mathcal{H}}}(\cdot)$ and $K$ categories for $S_{\overline{\mathcal{H}}}(\cdot)$. Hence, two detectors $G_{\widehat{\mathcal{H}}}(\cdot)$ and $G_{\overline{\mathcal{H}}}(\cdot)$ are acquired with the thresholds $\gamma_{\widehat{\mathcal{H}}}$ and $\gamma_{\overline{\mathcal{H}}}$, able to detect $\mathbf{X}_{out}$ with $\text{FPR}_{\widehat{\mathcal{H}}}$ and $\text{FPR}_{\overline{\mathcal{H}}}$, respectively. As shown in the Figure 2a, when $\text{FPR}_{\widehat{\mathcal{H}}} = \text{FPR}_{\overline{\mathcal{H}}}$, $A - B$ contains OOD samples exposed to $G_{\widehat{\mathcal{H}}}(\cdot)$ while hidden from $G_{\overline{\mathcal{H}}}(\cdot)$. Hence, we can collect hidden OOD samples from

$$\widehat{\mathbf{X}}_{out} = A - B, \ s.t. \ \text{FPR}_{\widehat{\mathcal{H}}} \leq \text{FPR}_{\overline{\mathcal{H}}}.$$

Taking $\text{FPR}_{\widehat{\mathcal{H}}} = 0.8\%$, $\text{FPR}_{\overline{\mathcal{H}}} = 6\%$, CIFAR-100 (Krizhevsky & Hinton, 2009) as $\mathbf{X}_{in}$ and ImageNet (Deng et al., 2009) as $\mathbf{X}_P$, $\widehat{\mathbf{X}}_{out}$ with 2925 samples can be collected from the corresponding OOD dataset $\mathbf{X}_{out}$ in OpenOOD (Yang et al., 2022). Subsequently, we evaluate various competitive methods to detect these samples. It is noteworthy that hidden OOD samples exist in a widespread OOD distribution $\mathcal{P}_{out}$ and the construction of specific $\widehat{\mathbf{X}}_{out}$ aims to bring the over-fitting in OOD detection from obscurity to the surface. Results in Figure 2b, c show that all methods exhibit continuous deteriorating trends to detect $\widehat{\mathbf{X}}_{out}$ as the training progresses, underscoring the tangible threat posed by hidden OOD samples. On the other hand, the existence of hidden OOD samples fundamentally limits the performance of OOD detection methods based on ID representation.

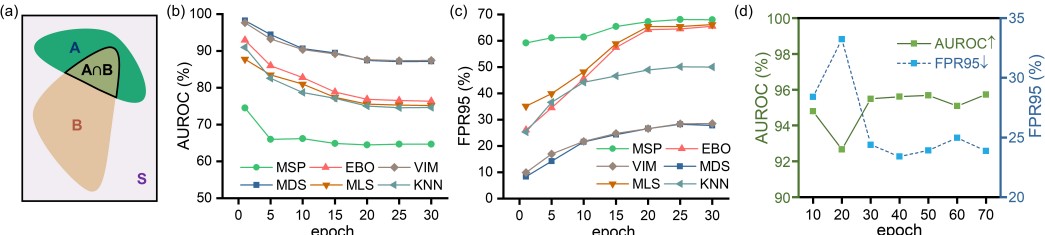

Figure 2: (a) The Venn diagram of OOD samples, where $S$ stands for general OOD samples in $\mathbf{X}_{out}$, $A$ and $B$ represent OOD samples that can be detected by $G_{\widehat{\mathcal{H}}}(\cdot)$ and $G_{\overline{\mathcal{H}}}(\cdot)$, respectively. The dynamics of AUROC (b) and FPR95 (c) of hidden OOD detection in the fine-tuning process of the pre-trained network with previous OOD detection methods and our ROMA (d).

## 4.2 ENCODER WITH MASKED IMAGE MODELING

For hidden OOD detection, $Encoder(\cdot)$ is expected to learn $\widehat{\mathcal{H}}$ for generalized representation. Outlier exposure seems to be a potential way to regularize $Encoder(\cdot)$ to learn $\widehat{\mathcal{H}}$. However, it will introduce external human efforts to find outliers(Katz-Samuels et al., 2022). Once there are ID samples in outliers, the outlier exposure will actually decrease the performance of ID recognition and OOD detection. Recent research (MOOD (Li et al., 2023)) found that reconstruction-based pretext tasks can provide a generally efficacious prior beneficial for the network in learning intrinsic data distributions, which significantly improves the network's performance in standard OOD detection. Inspired by this great success, we next present our framework to regularize $Encoder(\cdot)$ based on MAE.

With masked image modeling (MIM), MAE enables the encoder to learn $\mathcal{H}$ based on pixel-level understanding rather than patterns from classification. Thus, we utilize the identical architectures of $Encoder(\cdot)$ and $Decoder(\cdot)$ in MAE for feature extracting and reconstruction. Besides, an extra linear $Classifier(\cdot)$ is applied for classification. Both reconstruction and classification tasks share the same $Encoder(\cdot)$. Specifically, $Encoder(\cdot)$ is a ViT (Dosovitskiy et al., 2020), whose input is a subset of visible patches of the image $\mathbf{x}$. The $\mathbf{x}$ after tokenisation following previous setups (Arnab et al., 2021; Dosovitskiy et al., 2020) is $\mathbf{v} = Tokenise(\mathbf{x}) + \mathbf{p}$, where $\mathbf{p}$ denotes the positional embeddings. For $\mathbf{v} \in \mathbb{R}^{n \times d}$, $n$ is the total number of tokens and $d$ is the vector dimension. Random subsets of these tokens are masked with the ratio $\alpha$ and the rest unmasked tokens $\mathbf{u} = Mask(\mathbf{v}, \alpha)$ are processed by $Encoder(\cdot)$ with output $\mathbf{s} = Encoder(\mathbf{u})$, where $\mathbf{s} \in \mathbb{R}^{(1-\alpha) \cdot n \times d}$. For $Classifier(\cdot)$, the mask ratio $\alpha$ is $\alpha_C = 0$ and then $\mathbf{s}$ is averaged into feature $\mathbf{h} \in \mathbb{R}^d$ as the input. We have

$$\mathcal{L}_C(f_C(\mathbf{x}, \alpha_C), y) = \text{CrossEntropy}(Classifier(\mathbf{h}), y),$$

where $f_C(\cdot)$ represents the cascade operation of $Encoder(\cdot)$ and $Classifier(\cdot)$. For $Decoder(\cdot)$, the mask ratio $\alpha$ is $\alpha_M$ and the masked tokens $\mathbf{m} \in \mathbb{R}^{\alpha_M \cdot n \times d}$ are then inserted back into $\mathbf{s}$ whilst

adding new positional embeddings, by which we denote $\mathbf{z} = \text{Unshuffle}\,(\mathbf{s}, \mathbf{m}) \in \mathbb{R}^{n \times d}$ as the input. Finally, $Decoder(\cdot)$ processes $\mathbf{z}$ to reconstruct the original input $\mathbf{x}$ corresponding to the tokens in pixel space $\widetilde{\mathbf{x}}$. We have

$$\mathcal{L}_M(f_M(\mathbf{x}, \alpha_M), \mathbf{x}) = \mathbb{E}(\|Decoder(\mathbf{z}) - \widetilde{\mathbf{x}}\|^2),$$

where $f_M(\cdot)$ represents the cascade operation of $Encoder(\cdot)$ and $Decoder(\cdot)$.

### 4.3 REGULARIZATION FRAMEWORK

The two branches share the same $Encoder(\cdot)$. The feature $\mathbf{h}$ of the in-distribution image goes through a $Classifier(\cdot)$ followed by a Softmax function. In the reconstruction branch, there is a $Decoder(\cdot)$ whose input is the embedding $\mathbf{z}$ of the natural images.

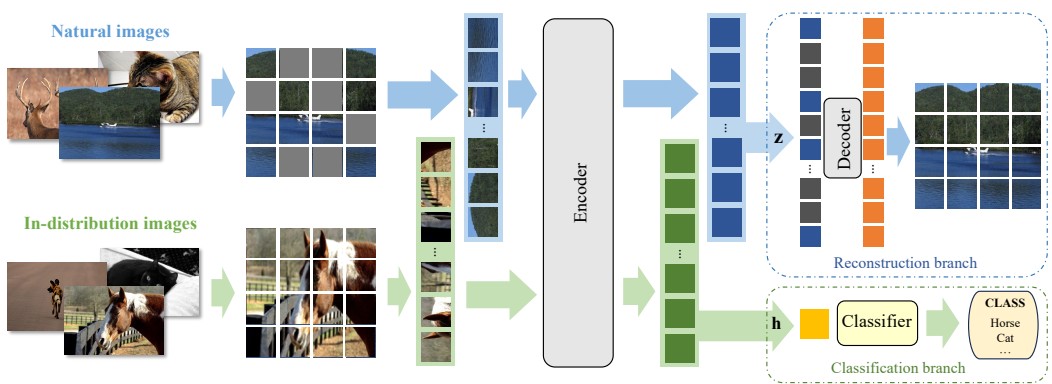

Figure 3: Overview of ROMA. The network includes two branches: reconstruction branch for natural images without considering distribution and the classification branch for ID images.

**Training procedure.** The overall training workflow consists of three steps: 1) pre-trains $Encoder(\cdot)$ in MIM task on large-scale pre-training dataset $\mathbf{X}_P$. 2) constructs data pair $(\mathbf{x}_{in}, \mathbf{x}_A)$ for input, where $\mathbf{x}_A$ is the auxiliary image randomly sampled from the candidate pool $\mathbf{X}_P$. 3) fine-tunes the two branches together with the encoder using $(\mathbf{x}_{in}, \mathbf{x}_A)$ whose target is $(y_{in}, \widetilde{\mathbf{x}}_A)$.

**Training objective.** The original training objective of normal classification is $\mathbb{E}_{(\mathbf{x},y) \sim \mathcal{D}_{in}}[\mathcal{L}(f(\mathbf{x}), y)]$, where $f(\cdot)$ is the network. To achieve superior performance in both hidden OOD and standard OOD detection, we propose ROMA with MIM regularization. As shown in Figure 3, the in-distribution images are fed into $Encoder(\cdot)$ and the feature vector $\mathbf{h}$ is sent to the classification branch. Meanwhile, self-supervised learning (SSL) of the auxiliary images is performed on the reconstruction branch, serving two purposes: 1) **regularization for ID features learning.** The network is only allowed to see part of the image space to realize correct reconstruction, reducing the over-fitting of ID representation learned for classification. 2) **regularization for generalized features learning.** ROMA regularizes the network to reconstruct masked auxiliary images instead of in-distribution images, encouraging the network to represent a more generalized feature space $\hat{\mathcal{H}}$. Thus, the training objective of ROMA is a combination of the learning objective for ID classification together with MIM regularization, which can be formulated as:

$$\mathbb{E}_{(\mathbf{x},y) \sim \mathcal{D}_{in}}[\mathcal{L}_C(f_C(\mathbf{x}, \alpha_C), y)] + \lambda \mathbb{E}_{\mathbf{x} \sim \mathbf{x}_A}[\mathcal{L}_M(f_M(\mathbf{x}, \alpha_M), \mathbf{x})].$$

**OOD inference.** During the testing procedure, only $Encoder(\cdot)$ and classification branch are retained. Since the regularization process focuses on the feature space $\mathcal{H}$, $S_{\mathcal{H}}(\cdot)$ based on Mahalanobis distance (Lee et al., 2018) is applied for OOD inference. Results in Figure 1c and Figure 2d show that ROMA alleviates the over-fitting in hidden OOD detection and there is a discernible trend indicating improved performance throughout the fine-tuning process. Further evaluation will be shown in the section 5.2.

## 5 EXPERIMENTS

In this section, we present extensive experiments to validate the superiority of ROMA, which improves standard OOD detection while effectively resisting hidden-OOD attacks. We provide comparisons

with previous competitive OOD detection methods and ROMA shows SOTA performance in most challenging tasks. Additionally, we perform ablation experiments to further elucidate ROMA's performance.

## 5.1 EXPERIMENTAL SETUP

**Datasets.** Following the standard OOD detection benchmarks based on OpenOOD (Yang et al., 2022), we use CIFAR-10, CIFAR-100, and ImageNet as ID datasets. For standard OOD detection, we use all far-OOD and near-OOD datasets corresponding to the selected ID datasets, which are summarized by OpenOOD. For hidden OOD detection, we use CIFAR-10, CIFAR-100, and BIMCV (Vayá et al., 2020) as ID datasets, and utilized our hidden-OOD finding strategy in the section 4.1 to find hidden OOD samples from their OOD datasets.

**Training details.** We evaluate numerous OOD detection methods with pre-trained networks, all of which are pre-trained on ImageNet (corresponding to $\mathbf{X}_P$). As OpenOOD concludes that vision transformer does not necessarily perform better than ResNet, we fine-tune two networks based on ResNet-18/50 (He et al., 2016) and ViT-B-16 (Dosovitskiy et al., 2020) in all benchmarks, and choose the one with better AUROC for OOD detection in the following evaluation. Appendix A.2 shows more details of training settings. For hyper-parameters of ROMA, we use $\lambda = 1$ and $\alpha_M = 0.75$.

**Evaluation metrics.** We evaluate the performance of OOD detection by measuring the following metrics: 1) the false positive rate (FPR95) of OOD examples when the true positive rate of in-distribution examples is 95%; 2) the area under the receiver operating characteristic curve (AUROC); and 3) the ID classification accuracy (ID-ACC).

## 5.2 RESULTS

**Hidden OOD detection benchmarks.** We first conduct hidden OOD detection benchmarks with competitive baseline methods, including MSP (Hendrycks & Gimpel, 2016), ODIN (Liang et al., 2018), MDS (Lee et al., 2018), EBO (Liu et al., 2020), MLS (Hendrycks et al., 2019a), ViM (Wang et al., 2022a), KNN (Sun et al., 2022), GEN (Liu et al., 2023), CIDER (Ming et al., 2022), OE (Hendrycks et al., 2019b), MOOD (Li et al., 2023), and ROMA. Among them, MOOD and ROMA require $\mathbf{X}_P$ to be incorporated into the subsequent training phase following pre-training, while OE necessitates the utilization of auxiliary images as outliers. To make a fair comparison with other *post-hoc* methods, auxiliary images used by MOOD, ROMA, and OE are randomly sampled from $\mathbf{X}_P$ and the amount of auxiliary images equals to the amount of ID images. As Table 1 shows, compared to directly using the pre-trained network for hidden OOD detection, the performance of all methods has declined after ID-ACC converged. This confirms that the learning of ID representations indeed causes some OOD samples to be hidden-to-detect. The results demonstrate that ROMA with effective regularization significantly outperforms other methods, achieving the best metrics of AUROC and FPR95. Although OE is widely regarded as an excellent regularization method for OOD detection, its performance is even worse than *post-hoc* methods when outliers contain in-distribution samples.

Table 1: Comparison between previous methods and ROMA in hidden OOD detection. PT represents the method of directly using the pre-trained ViT network and $S_{\mathcal{H}}(\cdot)$ for OOD detection. With the setting of $\mathrm{FPR}_{\widehat{\mathcal{H}}} = 0.8\%$, $\mathrm{FPR}_{\overline{\mathcal{H}}} = 6\%$, there are 173, 91, and 2925 hidden OOD samples collected for the ID datasets of BIMCV, CIFAR-10, and CIFAR-100, respectively.

| ID DATASET | | PT | MSP | ODIN | MDS | EBO | MLS | VIM | KNN | GEN | CIDER | LOGIT | OE | MOOD | ROMA |
|---|---|---|---|---|---|---|---|---|---|---|---|---|---|---|---|
| BIMCV | AUROC↑ | 99.98 | 83.52 | 99.11 | 92.52 | 83.52 | 83.52 | 92.13 | 98.48 | 83.52 | 58.27 | 23.03 | 73.22 | 99.84 | **99.97** |
| | FPR95↓ | 0.11 | 41.53 | 6.85 | 23.46 | 41.53 | 41.53 | 21.1 | 15.38 | 41.53 | 97.76 | 100 | 73.85 | 0.56 | **0** |
| | ID-ACC↑ | - | 73.06 | 73.06 | 73.06 | 73.06 | 73.06 | 73.06 | 73.06 | 73.06 | 73.06 | 67.79 | 63.64 | 73.4 | **75.42** |
| CIFAR-10 | AUROC↑ | 99.73 | 75.71 | 73.79 | 89.43 | 79.19 | 79.17 | 89.78 | 68.97 | 95.75 | 81.38 | 89.07 | 92.77 | 96.85 | **97.94** |
| | FPR95↓ | 0.8 | 49.7 | 79.14 | 23.95 | 53.6 | 53.6 | 22.37 | 68.39 | 50.95 | 52.32 | 31.04 | 30.81 | 19.59 | **8.74** |
| | ID-ACC↑ | - | 97.86 | 97.86 | 97.86 | 97.86 | 97.86 | 97.86 | 97.86 | 97.86 | 97.72 | 94.43 | 94.13 | 98.22 | **98.36** |
| CIFAR-100 | AUROC↑ | 99.87 | 64.7 | 60.75 | 87.13 | 76.35 | 75.14 | 87.39 | 74.67 | 63.83 | 71.31 | 72.4 | 75.25 | 91.83 | **96.91** |
| | FPR95↓ | 0.54 | 68.01 | 79.79 | 27.76 | 65.51 | 66.18 | 28.54 | 49.96 | 68.2 | 71.28 | 69.13 | 68.19 | 31.51 | **19.92** |
| | ID-ACC↑ | - | 85.79 | 85.79 | 85.79 | 85.79 | 85.79 | 85.79 | 85.79 | 85.79 | 85.72 | 75.84 | 74.78 | 85.06 | **85.8** |

**Standard OOD detection benchmarks.** We further evaluate the performance of ROMA in standard OOD detection. As shown in Table 2, ROMA still achieves SOTA performance with averaged AUROC/FPR95 of 91.27%/25.89%. For the most challenging task, near-OOD detection for ImageNet,

ROMA improves the SOTA AUROC/FPR95 from 77.03% (VIM)/66.99% (OE) to 79.23%/65.10%. The results are remarkable as ROMA is totally pre-trained and regularized with unlabeled images from ImageNet. Obviously, ROMA enables the network to detect more OOD samples that the methods learning $\bar{\mathcal{H}}$ for ID representation are blind to.

Table 2: Comparison between previous methods and ROMA. Full results are shown in Appendix A.3.

| | CIFAR-10 | | | | CIFAR-100 | | | | IMAGENET-1K | | | | AVERAGE | |
| | NEAR-OOD | | FAR-OOD | | NEAR-OOD | | FAR-OOD | | NEAR-OOD | | FAR-OOD | | | |
| | FPR95↓ | AUROC↑ | FPR95↓ | AUROC↑ | FPR95↓ | AUROC↑ | FPR95↓ | AUROC↑ | FPR95↓ | AUROC↑ | FPR95↓ | AUROC↑ | FPR95↓ | AUROC↑ |
|---|---|---|---|---|---|---|---|---|---|---|---|---|---|---|
| MSP | 53.58 | 87.69 | 31.23 | 91.03 | 54.55 | 80.38 | 58.89 | 77.65 | 81.85 | 73.52 | 51.69 | 86.04 | 53.29 | 83.17 |
| ODIN | 84.73 | 80.37 | 61.06 | 87.24 | 58.42 | 79.78 | 57.71 | 79.48 | 86.25 | 68.15 | 84.88 | 73.29 | 70.77 | 78.59 |
| MDS | 63.82 | 79.96 | 55.21 | 80.70 | 88.15 | 52.71 | 79.20 | 63.07 | 68.65 | 75.18 | 29.87 | 91.49 | 61.88 | 75.01 |
| EBO | 68.18 | 86.95 | 40.41 | 91.80 | 55.71 | 80.76 | 56.27 | 79.78 | 93.19 | 62.41 | 85.35 | 78.98 | 64.75 | 80.88 |
| MLS | 68.14 | 86.87 | 40.40 | 91.66 | 55.57 | 80.97 | 56.43 | 79.67 | 92.25 | 68.30 | 79.23 | 83.54 | 63.41 | 82.54 |
| VIM | 51.78 | 87.74 | 31.09 | 91.42 | 63.09 | 74.84 | 49.94 | 81.82 | 73.73 | 77.03 | 29.18 | **92.84** | 46.44 | 85.39 |
| KNN | 35.67 | 90.58 | 24.71 | 92.94 | 61.16 | 80.25 | 54.43 | 81.81 | 70.47 | 74.11 | 31.93 | 90.81 | 44.13 | 85.86 |
| GEN | 57.86 | 87.80 | 33.58 | 91.57 | 54.01 | 81.34 | 57.73 | 79.32 | 70.78 | 76.30 | 32.23 | 91.35 | 48.60 | 85.21 |
| CIDER | 31.36 | 90.7 | 19.17 | 95.05 | 72.02 | 73.10 | 54.22 | 80.49 | 71.69 | 68.97 | 28.69 | 92.18 | 42.82 | 84.84 |
| LOGIT | 28.00 | 92.62 | 12.65 | 97.03 | 64.91 | 77.76 | 50.02 | 80.62 | 68.56 | 74.62 | 31.32 | 91.54 | 40.98 | 86.55 |
| OE | 20.63 | 94.8 | 12.86 | 95.75 | **29.91** | 88.47 | 53.24 | 82.86 | 66.99 | 75.71 | 53.64 | 84.34 | 39.97 | 87.02 |
| MOOD | 15.00 | 96.65 | 2.18 | 99.25 | 40.42 | 89.14 | **23.15** | **90.55** | 73.39 | 72.06 | 35.41 | 88.78 | 28.65 | 90.18 |
| **OURS** | **13.01** | **97.32** | **1.64** | **99.47** | 34.46 | **89.81** | 28.24 | 87.30 | **65.10** | **79.23** | **26.46** | 92.06 | **25.89** | **91.27** |

**OOD detection with uneven distribution.** To assess ROMA's robustness, we examine the OOD detection performance when ID representation degrades as the unevenness of ID distribution increases. In this experiment, we modify the in-distribution unevenness of $\mathcal{D}_{in}$. Specifically, we leave one class unchanged and sample randomly 10% to 40% images from each of the rest nine classes to form an uneven ID dataset. This process is repeated ten times to generate ten uneven ID datasets and all methods are trained and evaluated on these datasets. The averaged performance is displayed in Table 3. Results show that metrics of both near-OOD and far-OOD detection degrade with the increasing unevenness of in-distribution. However, ROMA still outperforms other methods and has the minimal degradation. For the most challenging scenario with the sampling ratio of 10%, ROMA still maintains considerable AUROC/FPR95 of 95.13%/20.66% and 98.15%/5.28% in near-OOD and far-OOD detection respectively. Results further demonstrate effective regularization of ROMA for generalized features learning, which enables the network to maintain robust OOD detection even if ID representation has certain degradation.

Table 3: Performance of OOD detection with uneven training distribution. To form dataset with uneven in-distribution, we leave one class unchanged while randomly sampling images from each of the rest classes at the sampling ratio from 0.1 to 0.4. The original ID dataset is CIFAR-10.

| | RATIO 0.1 | | | | RATIO 0.2 | | | | RATIO 0.4 | | | |
| | NEAR-OOD | | FAR-OOD | | NEAR-OOD | | FAR-OOD | | NEAR-OOD | | FAR-OOD | |
| | FPR95↓ | AUROC↑ | FPR95↓ | AUROC↑ | FPR95↓ | AUROC↑ | FPR95↓ | AUROC↑ | FPR95↓ | AUROC↑ | FPR95↓ | AUROC↑ |
|---|---|---|---|---|---|---|---|---|---|---|---|---|
| MSP | 65.88 | 73.46 | 62.49 | 73.49 | 53.89 | 80.63 | 48.59 | 81.93 | 46.60 | 84.91 | 39.05 | 86.57 |
| ODIN | 66.78 | 75.87 | 57.39 | 79.14 | 59.27 | 81.18 | 47.59 | 84.90 | 60.16 | 83.28 | 45.67 | 87.62 |
| MDS | 89.20 | 55.45 | 79.82 | 61.62 | 85.29 | 59.25 | 75.14 | 66.81 | 75.49 | 67.77 | 63.49 | 73.04 |
| EBO | 64.06 | 77.25 | 60.17 | 76.79 | 54.64 | 82.71 | 48.35 | 83.78 | 51.25 | 85.96 | 42.75 | 87.49 |
| MLS | 64.05 | 77.09 | 60.27 | 76.65 | 54.62 | 82.61 | 48.43 | 83.69 | 51.25 | 85.89 | 42.79 | 87.43 |
| VIM | 66.86 | 75.79 | 58.29 | 78.28 | 56.81 | 80.67 | 45.76 | 84.83 | 49.79 | 84.67 | 36.88 | 88.28 |
| KNN | 87.71 | 40.56 | 82.59 | 44.39 | 68.77 | 74.85 | 57.65 | 79.63 | 47.60 | 85.59 | 37.13 | 88.14 |
| GEN | 65.69 | 73.81 | 62.59 | 73.90 | 54.12 | 80.49 | 49.07 | 81.68 | 47.24 | 84.85 | 39.80 | 86.44 |
| CIDER | 93.84 | 47.27 | 88.14 | 46.81 | 91.01 | 40.36 | 89.02 | 40.81 | 75.38 | 63.72 | 70.85 | 65.86 |
| LOGIT | 77.56 | 70.67 | 60.30 | 77.81 | 57.48 | 81.58 | 39.59 | 88.79 | 42.85 | 87.41 | 26.57 | 93.18 |
| OE | 66.85 | 74.83 | 64.68 | 66.19 | 50.64 | 83.52 | 47.22 | 76.98 | 33.89 | 90.54 | 29.08 | 88.35 |
| MOOD | 22.59 | 94.92 | 7.99 | 97.41 | 19.18 | 95.71 | 5.34 | 98.25 | 17.28 | 96.24 | 3.80 | 98.79 |
| **OURS** | **20.66** | **95.13** | **5.28** | **98.15** | **18.10** | **95.88** | **3.67** | **98.75** | **14.02** | **97.11** | **2.87** | **99.09** |

**Ablation study of ROMA.** In this section, we conduct ablations for ROMA in hidden OOD and standard OOD detection, where the ID dataset is CIFAR-100. For standard OOD detection, we show the average results of near-OOD and far-OOD detection. We first evaluate the effectiveness of designed training procedures (pre-trainning, fine-tuning, and self-supervised learning) for ROMA and results are shown in Table 4a. Compared with traditional fine-tuning procedure, ROMA improves the performance of both hidden OOD and standard OOD detection. We proceed to conduct ablation experiments varying the number $N_A$ of auxiliary images $\mathbf{x}_A$ (Table 4b) and the candidate dataset for SSL (Table 4c). Results demonstrate that increasing the number of auxiliary images within the training process for SSL does not consistently yield improved OOD detection metrics. Thus, it is

advisable to utilize an equal number of auxiliary images as the number ($N$) of ID images, striking a balance between computational resources and performance. Regarding the candidate dataset, ROMA with auxiliary images $\mathbf{x}_A$ sampled from ImageNet ($\mathbf{X}_P$) shows superior performance in hidden OOD detection as it has broad distribution for more generalized features learning. Notably, ROMA with $\mathbf{x}_A$ exclusively sampled from in-distribution (training) images achieves the best performance in standard OOD detection, underscoring the inherent advantage of the MIM task itself for OOD detection and the blind spots for standard OOD detection.

Table 4: Results of ablation study on (a) MIM pre-training (PT), fine-tuning (FT), and self-supervised learning (SSL); (b) the number ($N_A$) of auxiliary images; (c) candidate dataset for SSL.

(a) Ablations on PT, FT, and SSL.

| TASK | PT | FT | SSL | FPR95↓ | AUROC↑ |
|---|---|---|---|---|---|
| HIDDEN | √ | × | × | 3.32 | 98.73 |
| | √ | √ | × | 33.88 | 91.8 |
| | × | √ | √ | 92.73 | 51.87 |
| | √ | √ | √ | 19.92 | 96.91 |
| STANDARD | √ | × | × | 20.60 | 89.70 |
| | √ | √ | × | 37.29 | 88.20 |
| | × | √ | √ | 77.69 | 63.09 |
| | √ | √ | √ | 30.31 | 88.14 |

(b) Ablations on $N_A$.

| TASK | $N_A$ | FPR95↓ | AUROC↑ |
|---|---|---|---|
| HIDDEN | $0.5 \times N$ | 37.47 | 87.4 |
| | $1.0 \times N$ | 19.92 | 96.91 |
| | $2.0 \times N$ | 40.57 | 85.19 |
| | $5.0 \times N$ | 41.37 | 84.42 |
| STANDARD | $0.5 \times N$ | 30.31 | 90.01 |
| | $1.0 \times N$ | 30.31 | 88.14 |
| | $2.0 \times N$ | 32.08 | 89.31 |
| | $5.0 \times N$ | 32.33 | 89.13 |

(c) Ablations on the SSL dataset.

| TASK | SSL DATASET | FPR95↓ | AUROC↑ |
|---|---|---|---|
| HIDDEN | IMAGENET-1K | 19.92 | 96.91 |
| | PLACES365 | 45.84 | 80.95 |
| | CIFAR-10 | 35.94 | 91.77 |
| | IN-DISTRIBUTION | 33.53 | 92.53 |
| STANDARD | IMAGENET-1K | 30.31 | 88.14 |
| | PLACES365 | 37.15 | 85.58 |
| | CIFAR-10 | 26.54 | 90.98 |
| | IN-DISTRIBUTION | 26.32 | 90.99 |

## 5.3 DISCUSSION

With the reasoning ability learned from the MIM task in pre-training step and reconstruction branch, ROMA is actually able to process masked images on the classification branch. Specifically, $\alpha_C$ in the training procedure can be set to a non-zero value and each ID image will be randomly masked in each training epoch. During the testing procedure, the network is allowed to see the whole ID image and there is no masked patch, corresponding to $\alpha_C = 0$. Under this condition, adversarial training (ADV) with one step towards ID-supervised learning and one step backwards self-supervised learning may be more appropriate than directly sum up the weighted loss (SUM) of $\mathbb{E}_{(\mathbf{x},y)\sim\mathcal{D}_{in}}[\mathcal{L}_C(f_C(\mathbf{x}, \alpha_C), y)]$ and $\mathbb{E}_{\mathbf{x}\sim\mathbf{x}_A}[\mathcal{L}_M(f_M(\mathbf{x}, \alpha_M), \mathbf{x})]$. As shown in Figure 4, we deploy ROMA with $\alpha_C = 0.75$ in hidden OOD detection of CIFAR-100. The results demonstrate that this masking operation will further deepen the regularization of ID representation and reduce the computing resources required for training. The price in exchange is a reduction of arround 5% in ID-ACC. Overall, ROMA showcases more possibilities of MIM-based regularization so that trade-offs can be taken among the performance in ID recognition, OOD detection, and computing resources.

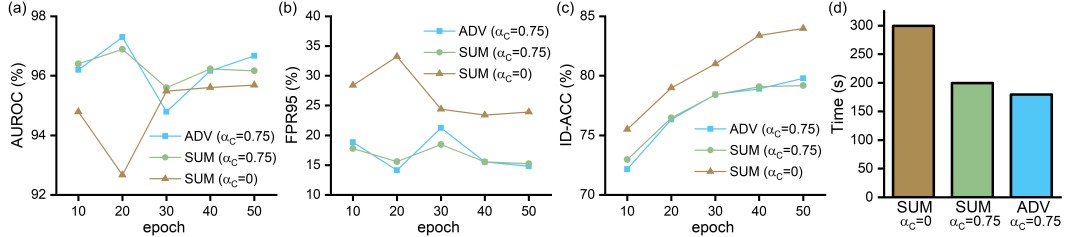

Figure 4: Results for ROMA fine-tuning with $\alpha_C = 0.75$ v.s. $\alpha_C = 0$. We show the (a) AUROC, (b) FPR95, (c) ID-ACC, and (d) averaged running time of an epoch for ROMA in hidden OOD detection.

## 6 CONCLUSION

In this paper, we introduce a task of hidden OOD detection, which exhibits the over-fitting of ID representation in OOD detection. To alleviate this problem, we propose a regularization framework for OOD detection with masked autoencoders (ROMA), utilizing masked image modeling to regularize the network to learn generalized features. With high proportion of masking, ROMA enables the network to perform well in both hidden OOD and standard OOD detection. This method can be easily adopted in practical settings as it can be applied with unlabeled auxiliary images regardless of distribution. We hope that our insights inspire future research to explore methods for OOD detection considering hidden OOD samples, which are critical complements to the standard OOD detection.

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

# A APPENDIX

The supplementary material serves as a repository for additional content, including algorithm, experimental configuration, detailed results, etc. These valuable assets are not be accommodated in the main paper due to page limitations.

## A.1 ALGORITHM

The algorithm of ROMA (includes pre-training, fine-tuning, and inferring) is shown in Algorithm 1.

---

**Algorithm 1** ROMA-based Out-of-distribution Detection Algorithm

---

**Require:** Large-scale set $\mathbf{X}_P$ for pre-training, in-distribution set $\mathbf{X}_{train}$ for fine-tuning, auxiliary set $\mathbf{X}_A$ for SSL, test set $\mathbf{X}_{test}$ for inferring, masking ratio $\alpha$, required True Positive Rate $\eta\%$
**Ensure:** Is $\mathbf{x}_{test} \in \mathbf{X}_{test}$ OOD or not?
 1: Pre-train $f_{mae}$ on $\mathbf{X}_P$ by minimizing

$$\mathcal{L}_M(f_{MAE}(\mathbf{x}_P, \alpha_M), \mathbf{x}_P)$$

 2: Fine-tune the $Encoder(\cdot)$ of $f_{mae}$ on $\mathbf{X}_{train}$ and $\mathbf{X}_A$ by minimizing

$$\mathbb{E}_{(\mathbf{x},y) \sim \mathcal{D}_{in}}[\mathcal{L}_C(f_C(\mathbf{x}, \alpha_C), y)] + \lambda \mathbb{E}_{\mathbf{x} \sim \mathbf{X}_A}[\mathcal{L}_M(f_M(\mathbf{x}, \alpha_M), \mathbf{x})]$$

 3: $\mathbf{h}_0 = Encoder(\mathbf{x}, \alpha_C = 0)$ for $\mathbf{x} \in \mathbf{X}_{train} \cup \mathbf{X}_{test}$
 4: Use $\mathbf{h}_0$ to calculate $S(\mathbf{x}_{test})$ for $\mathbf{x}_{test} \in \mathbf{X}_{test}$ and $S(\mathbf{x}_{train})$ for $\mathbf{x}_{train} \in \mathbf{X}_{train}$, where $S(\cdot)$ is defined by
$$S(\mathbf{x}) = -\min_i (\mathbf{h}_0 - \boldsymbol{\mu}_i)^T \boldsymbol{\Sigma}^{-1} (\mathbf{h}_0 - \boldsymbol{\mu}_i)$$
where $\boldsymbol{\mu}$ and $\boldsymbol{\Sigma}$ are the mean and covariance of $\mathbf{h}_0$ of the in-distribution set $\mathbf{X}_{train}$
 5: Compute threshold $\gamma$ as the $\eta$ percentile of $S(\mathbf{x}_{train})$.
 6: **if** $S(\mathbf{x}_{test}) > \gamma$ **then**
 7:     $\mathbf{x}_{test}$ is OOD
 8: **end if**

---

## A.2 EXPERIMENTAL CONFIGURATION

We directly employ the pre-trained network released by MAE (He et al., 2022a) including a ViT-B-16 as encoder, a relatively small ViT as decoder and a classifier head corresponding to ID classes. During the training process, we follow MAE and represent the image as a sequence of discrete tokens obtained by an image tokenizer. We randomly crop and resize images in both training datasets and auxiliary images to 224 × 224. Then we split each 224 × 224 image into a 14 × 14 grid of image patches, where each patch is 16 × 16. The patches are linearly-connected and input to the ViT-based encoder. After that, the generated embeddings of ID samples and auxiliary samples are then passed through classifier for classification and decoder for masked image reconstruction, respectively. Our augmentation policy includes random resized cropping, horizontal flipping, and color jittering. We train ROMA on CIFAR-10, CIFAR-100, and hidden-OOD datasets using 1 × NVIDIA RTX 3090 GPU and on ImageNet-1k using 8 × NVIDIA RTX 3090 GPU. More configuration details in the experiments are shown in Table 5 and Table 6.

Table 5: Parameters of the MAE.

| | ENCODER | | | DECODER | | | CLASSIFIER |
| BACKBONE | EMBED DIM | DEPTH | HEAD NUMBER | EMBED DIM | DEPTH | HEAD NUMBER | MLP RATIO |
|---|---|---|---|---|---|---|---|
| VIT-BASE | 768 | 12 | 12 | 512 | 8 | 16 | 4 |

Table 6: Configuration of the training parameters.

| LEARNING RATE | WARMUP EPOCHS | EPOCHS | ACCUM ITER | LAYER DECAY | DROP PATH | WEIGHT DECAY | BATCH SIZE |
|---|---|---|---|---|---|---|---|
| 5E-4 | 5 | 100 | 4 | 0.65 | 0.1 | 0.05 | 128 |

## A.3 DETAILED RESULTS FOR STANDARD OPENOOD BENCHMARKS

OpenOOD (Yang et al., 2022) subdivides standard OOD detection into two tasks: near-OOD and far-OOD detection. For benchmarks of CIFAR-10 shown in Table 7, (ID, OOD) pairs of (CIFAR-10, CIFAR-100) and (CIFAR-10, TinyImageNet) are used for near-OOD detection; (ID, OOD) pairs of (CIFAR-10, MNIST), (CIFAR-10, SVHN), (CIFAR-10, Texture), and (CIFAR-10, Places365) are used for far-OOD detection. For benchmarks of CIFAR-100 shown in Table 8, (ID, OOD) pairs of (CIFAR-100, CIFAR-10) and (CIFAR-100, TinyImageNet) are used for near-OOD detection; (ID, OOD) pairs of (CIFAR-100, MNIST), (CIFAR-100, SVHN), (CIFAR-100, Texture), and (CIFAR-100, Places365) are used for far-OOD detection. For benchmarks of ImageNet-1K shown in Table 9, (ID, OOD) pairs of (ImageNet-1K, SSB-Hard) and (ImageNet-1K, NINCO) are used for near-OOD detection; (ID, OOD) pairs of (ImageNet-1K, iNaturalist), (ImageNet-1K, Texture), and (ImageNet-1K, OpenImage-O) are used for far-OOD detection.

Table 7: Detailed performance on CIFAR-10.

| | CIFAR-100 | | TINYIMAGENET | | MNIST | | SVHN | | TEXTURE | | PLACES365 | |
|---|---|---|---|---|---|---|---|---|---|---|---|---|
| | FPR95↓ | AUROC↑ | FPR95↓ | AUROC↑ | FPR95↓ | AUROC↑ | FPR95↓ | AUROC↑ | FPR95↓ | AUROC↑ | FPR95↓ | AUROC↑ |
| MSP | 59.82 | 86.73 | 47.33 | 88.64 | 19.22 | 93.95 | 23.82 | 91.68 | 40.20 | 89.13 | 41.67 | 89.35 |
| ODIN | 84.87 | 79.55 | 84.59 | 81.18 | 15.39 | 96.60 | 69.04 | 84.62 | 83.19 | 83.86 | 76.62 | 83.87 |
| MDS | 66.33 | 79.47 | 61.30 | 80.44 | 74.73 | 65.90 | 39.63 | 88.67 | 35.98 | 91.43 | 70.50 | 76.79 |
| EBO | 72.69 | 85.55 | 63.68 | 88.35 | 15.49 | 96.32 | 29.34 | 92.60 | 60.43 | 88.63 | 56.38 | 89.63 |
| MLS | 72.66 | 85.49 | 63.63 | 88.26 | 15.53 | 96.12 | 29.26 | 92.46 | 60.44 | 88.55 | 56.38 | 89.51 |
| VIM | 57.11 | 86.67 | 46.46 | 88.80 | 30.79 | 89.50 | 21.57 | 93.56 | 22.13 | 94.66 | 49.89 | 87.95 |
| KNN | 39.19 | 89.60 | 32.14 | 91.57 | 21.72 | 94.11 | 21.62 | 92.85 | 24.93 | 92.99 | 30.57 | 91.82 |
| GEN | 63.63 | 86.71 | 52.09 | 88.89 | 18.01 | 95.00 | 24.08 | 92.18 | 46.12 | 89.36 | 46.12 | 89.74 |
| CIDER | 34.98 | 89.39 | 27.74 | 92.01 | 19.11 | 94.70 | 8.21 | 98.31 | 24.02 | 93.86 | 25.34 | 93.35 |
| LOGIT | 32.92 | 91.28 | 23.08 | 93.97 | 3.58 | 99.24 | 5.00 | 98.98 | 20.70 | 95.21 | 21.31 | 94.70 |
| OE | 33.68 | 91.50 | 7.59 | 98.10 | 28.66 | 88.60 | 2.16 | 99.31 | 8.33 | 97.97 | 12.31 | 97.13 |
| MOOD | 28.84 | 93.56 | 1.16 | 99.75 | 1.44 | 99.72 | 7.29 | 97.28 | 0.00 | 100.00 | 0.00 | 99.99 |
| OURS | 24.56 | 94.96 | 1.47 | 99.68 | 1.97 | 99.55 | 4.32 | 98.39 | 0.01 | 99.99 | 0.27 | 99.94 |

Table 8: Detailed performance on CIFAR-100.

| | CIFAR-10 | | TINYIMAGENET | | MNIST | | SVHN | | TEXTURE | | PLACES365 | |
|---|---|---|---|---|---|---|---|---|---|---|---|---|
| | FPR95↓ | AUROC↑ | FPR95↓ | AUROC↑ | FPR95↓ | AUROC↑ | FPR95↓ | AUROC↑ | FPR95↓ | AUROC↑ | FPR95↓ | AUROC↑ |
| MSP | 59.12 | 78.54 | 49.98 | 82.22 | 63.47 | 73.54 | 55.44 | 79.37 | 61.24 | 78.07 | 55.40 | 79.61 |
| ODIN | 60.98 | 78.03 | 55.86 | 81.53 | 50.36 | 82.09 | 62.16 | 75.56 | 59.83 | 80.46 | 58.51 | 79.80 |
| MDS | 92.08 | 49.56 | 84.22 | 55.86 | 75.04 | 61.77 | 75.78 | 63.40 | 79.88 | 71.33 | 86.09 | 55.78 |
| EBO | 58.88 | 79.01 | 52.53 | 82.51 | 57.57 | 77.30 | 50.84 | 82.67 | 60.21 | 79.33 | 56.47 | 79.81 |
| MLS | 58.93 | 79.20 | 52.21 | 82.74 | 57.93 | 76.91 | 51.14 | 82.33 | 60.24 | 79.36 | 56.38 | 80.06 |
| VIM | 71.20 | 71.70 | 54.99 | 77.98 | 48.36 | 80.94 | 44.73 | 83.87 | 46.26 | 86.39 | 60.41 | 76.09 |
| KNN | 73.31 | 77.00 | 49.00 | 83.49 | 55.51 | 79.44 | 50.38 | 83.87 | 52.57 | 84.14 | 59.27 | 79.78 |
| GEN | 58.56 | 79.39 | 49.44 | 83.28 | 61.42 | 75.66 | 53.68 | 81.53 | 60.78 | 79.53 | 55.03 | 80.57 |
| CIDER | 82.71 | 67.55 | 61.44 | 78.65 | 75.32 | 68.14 | 17.82 | 97.17 | 54.43 | 82.21 | 69.30 | 74.43 |
| LOGIT | 77.83 | 72.84 | 51.99 | 82.67 | 51.56 | 83.11 | 40.73 | 85.54 | 77.67 | 73.41 | 54.11 | 80.43 |
| OE | 59.69 | 77.03 | 0.13 | 99.90 | 58.28 | 79.24 | 38.94 | 92.20 | 58.87 | 80.65 | 56.87 | 79.35 |
| MOOD | 63.33 | 81.42 | 17.50 | 96.86 | 67.34 | 68.41 | 14.26 | 95.81 | 0.21 | 99.85 | 10.78 | 98.11 |
| OURS | 52.53 | 82.73 | 16.38 | 96.88 | 86.73 | 58.72 | 23.22 | 91.28 | 0.63 | 99.74 | 2.36 | 99.46 |

## A.4 DETAILED RESULTS FOR TOY DATASETS OF HIDDEN OOD DETECTION

Based on the same pre-trained network, we evaluate both our ROMA and traditional fine-tuning on the eight toy datasets. The detailed metrics are listed in Table 10. In the first table, three (ID, hidden OOD) pairs are collected from ImageNet-30. In the second table, the first two (ID, hidden OOD) pairs are collected from ImageNet-30, and the last one is selected from ImageNet-10. In the third table, the first (ID, hidden OOD) pair is collected from ImageNet-10, and the other is selected from CIFAR-100. As shown in Table 11, the average results show that the traditional fine-tuning procedure

leads to a degradation (from AUROC/FPR95 of 87.51%/40.80% to 85.18%/49.76%) in hidden OOD detection while ROMA improves the metrics to 90.31%/29.84%.

Table 9: Detailed performance on ImageNet-1K.

| | SSB-HARD | | NINCO | | INATURALIST | | TEXTURES | | OPENIMAGE-O | |
|---|---|---|---|---|---|---|---|---|---|---|
| | FPR95↓ | AUROC↑ | FPR95↓ | AUROC↑ | FPR95↓ | AUROC↑ | FPR95↓ | AUROC↑ | FPR95↓ | AUROC↑ |
| MSP | 86.41 | 68.94 | 77.28 | 78.11 | 42.40 | 88.19 | 56.46 | 85.06 | 56.19 | 84.86 |
| ODIN | 86.45 | 66.94 | 86.05 | 69.35 | 79.35 | 78.1 | 85.46 | 72.86 | 89.82 | 68.93 |
| MDS | 83.6 | 68.61 | 53.71 | 81.75 | 21.56 | 93.59 | 37.49 | 89.81 | 30.56 | 91.05 |
| EBO | 92.24 | 58.80 | 94.14 | 66.02 | 83.56 | 79.30 | 83.66 | 81.17 | 88.82 | 76.48 |
| MLS | 91.52 | 64.20 | 92.97 | 72.40 | 72.94 | 85.29 | 78.94 | 83.74 | 85.82 | 81.60 |
| VIM | 90.04 | 69.42 | 57.41 | 84.64 | 17.59 | 95.72 | 40.35 | 90.61 | 29.61 | 92.18 |
| KNN | 86.22 | 65.98 | 54.73 | 82.25 | 27.75 | 91.46 | 33.23 | 91.12 | 34.82 | 89.86 |
| GEN | 82.23 | 70.09 | 59.33 | 82.51 | 22.92 | 93.54 | 38.30 | 90.23 | 35.47 | 90.27 |
| CIDER | 82.69 | 59.34 | 56.46 | 78.60 | 28.70 | 90.76 | 20.86 | 96.38 | 36.52 | 89.39 |
| LOGIT | 82.08 | 65.70 | 55.04 | 81.73 | 20.75 | 94.57 | 40.82 | 89.30 | 32.38 | 90.75 |
| OE | 74.91 | 71.76 | 59.08 | 79.66 | 47.25 | 87.26 | 59.38 | 82.56 | 54.31 | 83.21 |
| MOOD | 86.93 | 64.12 | 59.85 | 80.00 | 31.49 | 88.13 | 33.50 | 91.17 | 41.24 | 87.03 |
| OURS | 80.85 | 73.17 | 49.35 | 85.28 | 17.62 | 94.04 | 36.10 | 89.64 | 25.65 | 92.50 |

Table 10: Detailed performance in toy datasets of hidden OOD detection comparing our ROMA with traditional fine-tuning.

| | ID: AMBULANCE, STRAWBERRY HIDDEN-OOD: TANK | | | | ID: CLOCK, AMBULANCE HIDDEN-OOD: PARKING METER | | | | ID: AIRLINER, TOASTER HIDDEN-OOD: TANK | | | |
|---|---|---|---|---|---|---|---|---|---|---|---|---|
| | ROMA | | FINETUNE | | ROMA | | FINETUNE | | ROMA | | FINETUNE | |
| EPOCH | FPR95↓ | AUROC↑ | FPR95↓ | AUROC↑ | FPR95↓ | AUROC↑ | FPR95↓ | AUROC↑ | FPR95↓ | AUROC↑ | FPR95↓ | AUROC↑ |
| 0 | 0.220 | 0.962 | 0.220 | 0.962 | 0.430 | 0.933 | 0.430 | 0.933 | 0.190 | 0.962 | 0.190 | 0.962 |
| 5 | 0.220 | 0.957 | 0.570 | 0.895 | 0.440 | 0.897 | 0.890 | 0.648 | 0.040 | 0.986 | 0.090 | 0.973 |
| 10 | 0.260 | 0.958 | 0.760 | 0.882 | 0.140 | 0.972 | 0.740 | 0.815 | 0.060 | 0.982 | 0.150 | 0.956 |
| 15 | 0.140 | 0.973 | 0.520 | 0.914 | 0.130 | 0.973 | 0.650 | 0.856 | 0.120 | 0.974 | 0.130 | 0.960 |
| 20 | 0.130 | 0.974 | 0.520 | 0.921 | 0.110 | 0.973 | 0.650 | 0.861 | 0.100 | 0.974 | 0.130 | 0.950 |
| 25 | 0.130 | 0.975 | 0.520 | 0.919 | 0.120 | 0.975 | 0.580 | 0.877 | 0.090 | 0.977 | 0.130 | 0.960 |
| 30 | 0.120 | 0.975 | 0.460 | 0.930 | 0.120 | 0.975 | 0.610 | 0.871 | 0.070 | 0.981 | 0.100 | 0.969 |

| | ID: FORKLIFT, REVOLVER HIDDEN-OOD: SNOWMOBILE | | | | ID: DRAGONFLY, TANK HIDDEN-OOD: FORKLIFT | | | | ID: AUTOMOBILE, CAT HIDDEN-OOD: TRUCK | | | |
|---|---|---|---|---|---|---|---|---|---|---|---|---|
| | ROMA | | FINETUNE | | ROMA | | FINETUNE | | ROMA | | FINETUNE | |
| EPOCH | FPR95↓ | AUROC↑ | FPR95↓ | AUROC↑ | FPR95↓ | AUROC↑ | FPR95↓ | AUROC↑ | FPR95↓ | AUROC↑ | FPR95↓ | AUROC↑ |
| 0 | 0.160 | 0.967 | 0.160 | 0.967 | 0.190 | 0.962 | 0.190 | 0.962 | 0.809 | 0.776 | 0.809 | 0.776 |
| 5 | 0.060 | 0.983 | 0.300 | 0.938 | 0.060 | 0.986 | 0.240 | 0.955 | 0.829 | 0.799 | 0.835 | 0.792 |
| 10 | 0.030 | 0.981 | 0.150 | 0.961 | 0.040 | 0.988 | 0.250 | 0.949 | 0.861 | 0.787 | 0.824 | 0.817 |
| 15 | 0.040 | 0.986 | 0.080 | 0.974 | 0.070 | 0.985 | 0.320 | 0.938 | 0.816 | 0.808 | 0.835 | 0.794 |
| 20 | 0.060 | 0.983 | 0.110 | 0.968 | 0.050 | 0.985 | 0.440 | 0.926 | 0.857 | 0.783 | 0.853 | 0.784 |
| 25 | 0.020 | 0.989 | 0.110 | 0.971 | 0.060 | 0.981 | 0.390 | 0.927 | 0.815 | 0.818 | 0.838 | 0.789 |
| 30 | 0.030 | 0.988 | 0.130 | 0.965 | 0.040 | 0.983 | 0.430 | 0.922 | 0.832 | 0.800 | 0.858 | 0.757 |

| | ID: AIRPLANE, CAT HIDDEN-OOD: SHIP | | | | ID: WILLOW TREE, APPLE HIDDEN-OOD: PINE TREE | | | |
|---|---|---|---|---|---|---|---|---|
| | ROMA | | FINETUNE | | ROMA | | FINETUNE | |
| EPOCH | FPR95↓ | AUROC↑ | FPR95↓ | AUROC↑ | FPR95↓ | AUROC↑ | FPR95↓ | AUROC↑ |
| 0 | 0.465 | 0.859 | 0.465 | 0.859 | 0.800 | 0.579 | 0.800 | 0.579 |
| 5 | 0.459 | 0.886 | 0.679 | 0.821 | 0.800 | 0.652 | 0.750 | 0.615 |
| 10 | 0.510 | 0.875 | 0.637 | 0.826 | 0.840 | 0.607 | 0.840 | 0.626 |
| 15 | 0.371 | 0.913 | 0.561 | 0.848 | 0.740 | 0.658 | 0.830 | 0.635 |
| 20 | 0.385 | 0.907 | 0.587 | 0.849 | 0.760 | 0.585 | 0.840 | 0.585 |
| 25 | 0.371 | 0.916 | 0.585 | 0.852 | 0.790 | 0.649 | 0.790 | 0.601 |
| 30 | 0.385 | 0.907 | 0.573 | 0.839 | 0.790 | 0.616 | 0.820 | 0.561 |

Table 11: Average results of hidden OOD detection in toy datasets comparing our ROMA with traditional fine-tuning.

| | ROMA | | FINETUNE | |
|---|---|---|---|---|
| EPOCH | FPR95↓ | AUROC↑ | FPR95↓ | AUROC↑ |
| 0 | 40.80 | 87.51 | 40.80 | 87.51 |
| 15 | 30.34 | 90.88 | 49.08 | 86.49 |
| 30 | 29.84 | 90.31 | 49.76 | 85.18 |

