# OpenReview forum: "ROMA: Regularization for Out-of-distribution Detection with Masked Autoencoders"
_ICLR.cc/2025/Conference — ICLR 2025 Conference Withdrawn Submission_

### Official Review · Reviewer_EWpZ · 2024-10-29

**Soundness:** 3
**Presentation:** 2
**Contribution:** 3
**Rating:** 5
**Confidence:** 3

**Summary:**

The paper introduces ROMA, a Regularization framework for Out-of-Distribution (OOD) Detection using Masked Autoencoders (MAE). Addressing the novel concept of “hidden OOD detection,” the authors propose that in-distribution (ID) representations alone may not be effective for detecting certain OOD data. ROMA utilizes masked image modeling as a regularization mechanism to enhance the generalization ability of the network by training it with unlabeled auxiliary images. Extensive experiments demonstrate that ROMA surpasses existing OOD detection methods in challenging hidden OOD and standard OOD detection benchmarks, showcasing significant robustness against overfitting.

**Strengths:**

1. Originality: The introduction of hidden OOD detection highlights a novel problem that conventional ID-focused methods may overlook, thus broadening the scope of OOD detection research.
2. Technical Depth: The use of masked autoencoders for regularization is an intelligent adaptation from NLP, creatively applied to image-based OOD detection tasks.
3. Experimental Validation: Comprehensive benchmarks, including challenging datasets and thorough comparison with baseline methods, validate ROMA’s effectiveness.
4. Practical Applicability: ROMA’s independence from labeled outlier data makes it adaptable for real-world applications where labeled auxiliary data is limited.

**Weaknesses:**

1. Clarity of the Task Definition: The term “hidden OOD” might benefit from further clarification, as it could be challenging for readers unfamiliar with OOD nuances. A more descriptive introduction, along with concrete examples, would improve accessibility.
2. Visual Explanation: Including a teaser figure or illustrative flowchart of the detection process could aid in visualizing how ROMA addresses the limitations of existing methods.

**Questions:**

1. For practical implementation, what are the computational costs associated with ROMA’s dual-branch architecture? How might this impact deployment in resource-constrained environments?

2. Can the authors provide additional guidance on tuning the hyperparameters for the auxiliary image masking ratio and the reconstruction loss weight?

---

### Official Review · Reviewer_Z88Q · 2024-10-30

**Soundness:** 2
**Presentation:** 2
**Contribution:** 2
**Rating:** 5
**Confidence:** 3

**Summary:**

The paper presents ROMA, a novel regularization framework for out-of-distribution (OOD) detection that leverages masked autoencoders (MAE) to address the limitations of in-distribution (ID) representations in detecting challenging OOD instances, particularly "hidden" OOD samples. Existing OOD methods often struggle with such hidden samples due to overfitting on ID representations, which can make certain OOD samples appear indistinguishable from ID data. ROMA tackles this by combining supervised classification on ID data with self-supervised masked image modeling on unlabeled auxiliary data, fostering more generalized feature learning. Extensive experiments demonstrate ROMA's superiority in both standard and hidden OOD benchmarks, showcasing state-of-the-art performance and robustness against ID overfitting. The authors highlight ROMA's practical applicability by avoiding reliance on explicit outlier datasets, making it well-suited for real-world OOD detection tasks.

**Strengths:**

Here are the strengths of this paper in my opinion:

1) Effective Regularization: Utilizes masked autoencoders (MAEs) to regularize feature learning, reducing overfitting on in-distribution (ID) data and enhancing robustness to challenging OOD samples.

2) Combination of Supervised and Self-Supervised Learning: Integrates supervised classification with self-supervised masked image modeling, promoting generalized feature learning without requiring labeled outlier data.

3) Strong Empirical Results: Demonstrates state-of-the-art performance across standard and hidden OOD benchmarks, validated through extensive experiments and ablation studies.

4) Practical Applicability: Avoids dependence on labeled outlier data, making ROMA more scalable and adaptable to real-world applications where labeled outliers are unavailable.

**Weaknesses:**

Here are the weaknesses of this paper in my opinion:

1) Limited Comparison with Other SOTA Methods: The paper does not consider several recent state-of-the-art (SOTA) methods, such as ASH [1], GradOrth [2], and GradNorm [3], which would strengthen the benchmark comparisons. Notably, ASH’s application of sparsity makes it particularly relevant for comparison, given ROMA’s use of masked autoencoders.

2) Clarity and Accessibility: The paper is challenging to follow in certain sections. For example, line 220 introduces terms A and B without defining them, which makes the explanation harder to understand. Similarly, more detail is needed to clarify comparisons involving metrics like FPR for different representations.

3) Complexity of Method Description: The dual-branch structure of ROMA, with supervised and self-supervised learning branches, could benefit from a clearer, step-by-step breakdown. The current description may overwhelm readers unfamiliar with masked autoencoder architectures or complex OOD detection setups.

4) Scalability in Resource-Constrained Settings: ROMA relies on auxiliary image sets and masked autoencoder operations, which could demand significant computational resources. Additional analysis of ROMA's efficiency in resource-constrained environments, or a comparison with lighter approaches, would provide more practical insights for broader applicability.

[1] Djurisic, Andrija, et al. "Extremely simple activation shaping for out-of-distribution detection." arXiv preprint arXiv:2209.09858 (2022).
[2] Behpour, Sima, et al. "GradOrth: a simple yet efficient out-of-distribution detection with orthogonal projection of gradients." Advances in Neural Information Processing Systems 36 (2024).
[3] Huang, Rui, Andrew Geng, and Yixuan Li. "On the importance of gradients for detecting distributional shifts in the wild." Advances in Neural Information Processing Systems 34 (2021): 677-689.

**Questions:**

Please address the weakness points.

---

### Official Review · Reviewer_zn6T · 2024-11-03

**Soundness:** 2
**Presentation:** 2
**Contribution:** 2
**Rating:** 3
**Confidence:** 4

**Summary:**

This paper introduces ROMA, a regularization framework for out-of-distribution (OOD) detection using Masked Autoencoders (MAE). ROMA addresses the challenge of hidden OOD detection, where certain OOD samples are insufficient for distingushing with ID representations. By employing a self-supervised masked image modeling task, ROMA enables the network to preserve generalized features, improving its ability to detect hidden OOD samples.

**Strengths:**

- This paper goes beyond the typical focus on far-OOD settings in OOD detection and addresses more challenging scenarios, such as hidden OOD detection, which is often overlooked but crucial for robust real-world applications.
- The paper introduces a novel approach, ROMA, which leverages masked image modeling as a regularization technique to prevent overfitting to ID representations, enabling the network to retain more generalized features crucial for detecting both standard and hidden OOD samples.

**Weaknesses:**

- Hidden OOD Detection's Novelty: The concept of Hidden OOD Detection proposed in this paper is not entirely new; it closely resembles the existing Near-OOD Detection task, which has already been well-defined in the field.
- Reframing Near-OOD as Hidden OOD: The Hidden OOD Detection defined in Section 3.2 is essentially a redefinition of Near-OOD detection using a different approach. Additionally, the paper does not propose a specialized benchmark setting tailored to this task in the experimental section.
- Growing Research on difficult OOD Detection: Recently, many approaches have been introduced to improve OOD detection beyond simple far-OOD tasks. Particularly, methods that utilize synthetic datasets without labeling costs have proven to be effective. It is crucial to compare ROMA against these baselines (e.g., VOS, NPOS, Dream-OOD, SONA).
- Limitations of Data-Driven Methods: The introduction should further elaborate on the limitations of data-driven methods for OOD detection, providing more context on the challenges they face and the importance of addressing these limitations.
- The additional method necessary on ROMA approach to directly capture the fine-grained semantic differences between ID and hidden OOD samples.

[1] Vos: Learning what you don't know by virtual outlier synthesis. \
[2] Non-Parametric Outlier Synthesis \
[3] Dream the Impossible: Outlier Imagination with Diffusion Models \
[4] Diffusion based Semantic Outlier Generation via Nuisance Awareness for Out-of-Distribution Detection

**Questions:**

see the weakness

---

### Official Review · Reviewer_ZmnJ · 2024-11-04

**Soundness:** 3
**Presentation:** 3
**Contribution:** 3
**Rating:** 5
**Confidence:** 3

**Summary:**

The paper presents a novel OOD detection task called hidden OOD detection, where in-distribution (ID) representations offer limited, or even counterproductive assistance in identifying hidden OOD data. This method employs MAE pre-trained by MIM pretext task as the basic network, regularizing the network to preserve generalized features against hidden OOD samples. Empirical results demonstrate that the suggested approach achieves promising performance on the hidden OOD detection task.

**Strengths:**

1. It is clearly written and well-organized.
2. The proposed methodology is straightforward and easy to understand. A comprehensive set of experiments support the method to a certain extent.

**Weaknesses:**

There are several potential weaknesses in the work:

1. The ROMA method itself does not seem to exhibit a high level of innovation. Additionally, the analysis of the advantages of the proposed method over previous approaches in the context of the new task appears to be somewhat ambiguous.
2. Regarding the research on the proposed hidden OOD detection task, there is a lack of detailed theoretical discussion on the concept and characteristics of hidden OOD in the task definition.

**Questions:**

1. Based on Table 1 and the ablation study, the effectiveness of the ROMA method appears to stem primarily from the MIM pre-training (PT) component. Further, related OOD literature has already noted the concept of reconstruction-based pretext tasks. Could the authors further clarify the distinctions between their work and previous studies, such as MOOD and MOODv2?
2. What is the distinction between the proposed task (hidden OOD detection) and the challenge of mining difficult OOD samples at the edges of the ID manifold?

---

### Note · Authors · 2024-12-10

I have read and agree with the venue's withdrawal policy on behalf of myself and my co-authors.